# Influence of the Addition of Rare Earth Elements on the Energy Storage and Optical Properties of Bi_0.5_Na_0.5_TiO_3_–0.06BaTiO_3_ Polycrystalline Thin Films

**DOI:** 10.3390/ma16062197

**Published:** 2023-03-09

**Authors:** Ilham Hamdi Alaoui, Mebarki Moussa, Nathalie Lemée, Françoise Le Marrec, Anna Cantaluppi, Delphine Favry, Abdelilah Lahmar

**Affiliations:** Laboratory of Condensed Matter Physics, University of Picardie Jules Verne, 33 Rue Saint Leu, 80039 Amiens, France

**Keywords:** BNT–BT thin films, rare earth addition, photoluminescence, dielectric measurements, energy storage

## Abstract

Rare earth element-doped Bi_0.5_Na_0.5_TiO_3_–BaTiO_3_ (BNT–BT–RE) polycrystalline thin films were processed on a platinized substrate by chemical solution deposition. The microstructure, dielectric, and ferroelectric properties were investigated for all prepared films. It was found that the incorporation of rare earth elements into the BNT–BT matrix increases both the dielectric constant and the breakdown strength while maintaining low dielectric losses, leading to an enhancement of the energy storage density to *W_rec_* = 12 and 16 J/cm^3^ under an effective field of *E* = 2500 kV/cm, for Nd- and Dy-based films, respectively. The optical properties of films containing the lanthanide element were investigated and the obtained results bear interest for luminescence applications. The simultaneous appearance of ferroelectric and optical properties in the system under investigation is very promising for advanced optoelectronic devices.

## 1. Introduction

Energy storage may take various forms, electrochemical, thermal, mechanical, etc. However, in electrical and electronic systems, energy storage requirements come into play in two fundamental ways: the first concerns the use of longer-term electrostatic discharges including energy supply in the case of redundant energy sources. The second is short-term storage, where components of electrically powered circuits store energy in an electrostatic or electromagnetic form [1]. Capacitors and inductors are typical examples. Recently, a revival of interest is devoted to lead-free dielectric capacitors for electrostatic energy storage [2,3,4,5,6,7,8,9,10,11]. Attracted by their ultra-fast charging–discharging characteristics combined with their high-power density, the industrial and research communities focus their efforts on chapping environmentally friendly materials able to meet the needs of modern technology [12]. Antiferroelectric materials are best suited for energy storage because they have high saturation polarization, low residual polarization, and low dielectric losses [13,14]. Unfortunately, most of these materials are lead-based, which is toxic and harmful to human health and the environment [15]. Alternatively, ferroelectric relaxors can also store a large amount of electrical energy because they have a slim hysteresis loop and a large saturation polarization, and a high coercive field [16,17,18,19,20,21].

Among the interesting lead-free perovskites, Bi_0.5_Na_0.5_TiO_3_ (abbreviated as BNT) has been paid a great deal of attention, owing to its excellent ferroelectric properties and also the presence of a likely antiferroelectric phase transition called the depolarization transition (*T_d_*). BNT was reported to exhibit an A-site disorder with the simultaneous presence of both Na and Bi elements [22]. It has been demonstrated in several works using advanced characterization tools such as X-ray and electron diffraction techniques, that at room temperature, BNT is considered to be of cubic symmetry with rhombohedral *R3c* and tetragonal *P4bm* polar nanoregions, albeit a dominating of *R3c* can be spotted at RT. However, in the temperature-dependent dielectric permittivity, a shoulder is observed around *T_d_*, which is promoted only in the case of substitution or polling, resulting from a mixed contribution from *R3c*-to-*P4bm* polar nanoregions transition [23]. Around this transition, BNT-based materials exhibit double-pinched ferroelectric hysteresis loops [2,4,5,23]. The origin of this behavior has been widely discussed in the literature [24,25].

Unfortunately, the pure BNT phase exhibits a large coercive field (*E_C_* = 73 kV/cm) as well as high dielectric losses leading to a small breakdown field, which limits its consideration for energy storage [1,5]. Thus, the compositional modulation by substitution in the pure BNT was found to be a good strategy for overcoming these drawbacks.

For instance, several BNT-based ceramic materials are reported to endorse interesting energy density (*W_rec_*). The focus was on how to increase the dielectric breakdown to increase the stored energy, although, the obtained energy remains below expectations for the technological applications. This is why particular interest has been directed toward thin films of these materials in which a high dielectric breakdown field can be reached [4]. The review of the literature shows that the energy stored in thin films greatly exceeds that of ceramics by many orders of magnitude [26,27,28]. Adjusting the composition close to the morphotropic boundary phase (MBP) of BNT-BaTiO_3_ (BNT–BT) was found to be an appropriate way to improve the stored energy. Yuan Yao et al. [9] reported a *W_rec_* = 32 J/cm^3^ for 0.9(0.94Bi_0.5_Na_0.5_TiO_3_-0.06BaTiO_3_)-0.1NaNbO_3_ thin films deposited on LaNiO_3_ using a radio-frequency (RF) magnetron sputtering technique. Further, an ultrahigh recoverable energy storage density of about 54 J/cm^3^ was reported for Mn-modified BNT–BT thin films prepared by chemical solution deposition [7]. In this perspective, we have investigated the addition of rare earth elements to this MBP composition to increase the stored energy and to also take advantage of the presence of lanthanide elements to highlight luminescence properties. The simultaneous occurrence of ferroelectric and optical properties in the studied system holds great promise for optoelectronic and photonic applications.

## 2. Materials and Methods

Bismuth acetate (III) (C_6_H_9_BiO_6_), Sodium acetate (C_2_H_3_NaO_2_), Barium acetate (Ba(C_2_H_3_O_2_)_2_), rare earth acetate RE(O_2_C_2_H_3_)_3_ (RE = Nd; Dy), and titanium isopropoxide (C_12_H_28_O_4_Ti) were used as starting materials to prepare a 0.3 mol/L precursor solution. Firstly, Bi, Ba, Na, and RE acetates were dissolved in acetic acid with appropriate stoichiometry. Separately, Ti-isopropoxide was dissolved in 2-Methoxyethanol and stabilized with acetylacetone (AcAc) selected as a chelating agent. Finally, the two solutions were mixed and stirred to yield a stable solution at room temperature. Thin film fabrication was performed by multiple-layer deposition using spin-coating onto Pt/SiN substrates. The pyrolysis of each layer was carried out on a hotplate at 400 °C for 10 min. The crystallization of the prepared films was carried out in a tube furnace at 600 °C under an O_2_ atmosphere for 30 min. The final thickness was estimated to be around 460 nm.

An FEI Quanta 200 FEG Environmental SEM was used to analyze the microstructure of the investigated thin films. The phase purity was checked by X-ray diffraction using a four-circle Bruker Discover Advance D8 diffractometer with CuKα = 1.5406 Å. Raman spectroscopy was performed using a Renishaw micro-Raman spectrometer under a green laser excitation wavelength of 514.5 nm. The dielectric investigations were conducted using a Solartron Impedance analyzer SI-12060. A ferroelectric test system (TF Analyzer 2000, aixACCT, Aachen, Germany) was used to collect P–E hysteresis loops. Finally, the luminescence properties were investigated using a LabRAM HR Evolution spectrophotometer under an excitation laser wavelength of 360 nm for the Dy phase and 786 nm for the Nd phase.

## 3. Results

### 3.1. Microstructural and Structural Investigations

Figure 1 shows scanning electron microscopy images for the investigated samples. The films were crack-free, dense, and fine-grained. As can be seen from Figure 1b,c, doping with rare earth elements induced a reduction in the grain size compared to the BNT–BT films. The average grain size in undoped BNT–BT was found to be 0.62 μm, which is much larger than that observed in doped films—0.38 μm and 0.36 μm for BNT–BT–Dy and BNT–BT–Nd, respectively. A similar behavior was reported by Yanjiang Xie et al. [7] while adding Mn to the BNT–BT polycrystalline thin films. They attributed this behavior to the “solute drag” effect. The authors surmised that the nonuniform distribution of the doping element into the host matrix leads to the reduction in the driving force for grain boundary migration. The inhibition of grain growth by the incorporation of rare earth elements has been reported in many types of research work [4]. A typical cross-section of one of these films is presented in Figure 1d, revealing a uniform thickness of about 463 nm. For doped films, the thickness was found to be around 472 nm and 475 nm for Dy and Nd-based films, respectively.

Figure 2 presents the EDX spectra for the compositional distribution in the investigated films. As expected, the characteristics of starting elements Bi, Na, Ba, Ti, and O, are detected for all samples. Further, the supplementary lanthanide element is detected in the doped films, which confirms the successful inclusion of Nd and Dy elements into the BNT-BT matrix.

Figure 3 shows the XRD diagrams of BNT-BT and lanthanide-doped BNT-BT films. The zoom on the reflection around the first order of the substrate clearly shows the existence of a splitting of the film’s peak on both sides of 40° (Figure 3a). These two peaks seem to be related to the film and not to the substrate. To obtain further information about this splitting, the grazing incidence mode (GI) was used to eliminate the substrate contribution, as shown in Figure 3b. The obtained XRD patterns were indexed based on the rhombohedral BNT and the tetragonal BT. The coexistence of rhombohedral *R3c* and tetragonal *P4bm* has already been highlighted in several works to be around *x* = 0.6 (MBP region) [29,30]. The splitting of the (202)_BNT_ peak (indexed in rhombohedral NBT) to two peaks (002)_BT_ and (200)_BT_ (indexed in tetragonal BaTiO_3_) and the combination of (003)_BNT_ and (021)_BNT_ into only one peak was reported as a signature of the complete transformation from the *R3c* phase to the *P4mb* phase. In this study, the splitting was achieved; but not the combination, which confirms the coexistence of both rhombohedral and tetragonal phases.

### 3.2. Dielectric Investigations

The frequency dependence of the dielectric constant and dielectric loss of BNT–BT and BNT-BT-RE (Dy and Nd) thin films at room temperature are plotted in Figure 4. The beneficial effect of the addition of lanthanide on the dielectric properties can be seen clearly. The dielectric constant increased from 600 for the BNT-BT phase to around 1000 for films containing rare earth elements; this behavior may be linked to the reduction in oxygen vacancies formed by the Bi volatility. In this case, the substitution of Bi by a nonvolatile lanthanide element should reduce the number of vacancies, according to the following equations (using Vink–Kröger notation):(1)$$2{\mathrm{Bi}}_{\mathrm{Bi}}+3O_{O}\to 2V_{\mathrm{Bi}}^{\prime \prime \prime }+3V_{\ddot{O}}+{\mathrm{Bi}}_{2}O_{3},$$
(2)RE2O3+2VBi‴+6h• →2REBi+32O2,

Figure 5 displays the thermal variation in the dielectric constant and dielectric loss for the investigated samples. No clear dielectric anomalies were depicted from the plot of BNT-BT and BNT-BT-Dy; only humps were observed. To highlight the possible phase transitions around these humps, the differential curves of the dielectric constant (∂Ɛr/∂T) as a function of temperature were plotted in the inset of Figure 5a. As can be seen from the plots, a visible anomaly is observed at around 120 °C that could be attributed to the depolarization temperature (*T_d_*). In contrast, the BNT–BT–Nd film shows one visible dielectric transition at *T_m_* = 270 °C.

A maximum around this temperature is observed for BNT-BT thin films with Na and/or Bi excess around the MBP composition [31]. It is worth mentioning that in bulk BNT-xBT solid solutions near the MBP, Badapanda et al. reported that the two observed anomalies (*T_d_* and *T_m_*) merged into one for the composition *x* = 0.8, then appeared again for *x* = 0.9. The authors have considered *x* = 0.8 as the highest MBP composition [32]. Figure 5b exhibits the frequency dependence of the thermal variation in the dielectric loss for BNT-BT. A clear relaxation phenomenon is observed that agrees well with the reported bibliography [23,24,26,29,31]. Note that similar behavior is observed for doped films (not shown here).

### 3.3. Ferroelectric and Energy Storage Investigations

Figure 6a shows the *P-E* hysteresis loop of the parent BNT-BT matrix of composition *x* = 0.06 located in the MBP region. The obtained *P-E* loop is characterized by a linear and slim shape. It is noted that the increase in the applied field does not induce polarization saturation; only an increase in the linearity of the hysteresis loop is observed. In a previous work [2,4], we found that the inclusion of lanthanide elements in the BNT matrix led to changes in the order of the A/B-site cations, which induced local random fields and thus a structural heterogeneity, useful for improving the energy storage properties.

Indeed, depending on the amount of RE doping into BNT (a non-ergodic relaxor material), different relaxation states could be obtained until a paraelectric-like state is achieved.

In this work, we surmise the coexistence of the ergodic and non-ergodic in BNT-BT caused by the *R3c* and *P4mb* nanopolar competition, which causes local random fields that break the long-range ferroelectric order instead of a short-range polar order. This is in good agreement with the X-ray diffraction investigation, where this competition is highlighted by the presence of rhombohedral and tetragonal diffraction peaks. Figure 6b shows the comparison of the *P-E* loops of undoped and doped BNT-BT thin films. The obtained slim hysteresis loop is a signature of the complete ergodic relaxor phase [4,5]. In fact, the substitution of Bi by Ba in the BNT matrix promotes the room-temperature relaxor properties, which is in good agreement with the result obtained in the dielectric investigations (Figure 5b). In addition, the introduction of the additional Bi substitution (inclusion of the RE elements) results in an increased disorder within the structure and, consequently, the relaxor behavior is amplified, leading to a decrease in the remanent polarization, and the hysteresis becomes slimmer.

The energy storage properties of the prepared thin films can be determined using the *P-E* loops by the integration of the polarization versus the electric field using the following equation:(3)$$W_{rec}=\int_{P_{r}}^{Pm}EdP,$$

The condition to obtain a high energy storage density is to have a high Δ*P* = *P_m_* − *P_r_* value as well as a high breakdown electric field value. Energy efficiency is an interesting parameter to evaluate the discharge property. It can be determined using the following equation:(4)$$\eta =\frac{W_{rec}}{W_{rec}+W_{loss}}\times 100,$$
where
(5)$$W_{loss}=W_{stored}-W_{rec},$$

It is noted that the condition of high energy efficiency simultaneously requires a high recovered energy and a low lost energy (Figure 7a).

The energy storage parameters were determined versus different values of the applied field in the positive part of the hysteresis loops (Figure 7b–d). Table 1 summarized the calculated parameter for the studied thin films in a comparison with reported data in the literature.

Figure 8 shows the variation in *W_rec_* and η as a function of the applied electric field for all investigated films. It can be seen from the plots that *W_rec_* increases with an increasing applied electric field. However, for BNT-BT films, it was not possible to apply a high electric field because of the breakdown effect. Contrarily, films containing elemental lanthanide exhibit high dielectric rigidity with a high breakdown electric field, which helps to increase the recovered energy density from 4.5 J/cm^3^ to around 6.5 J/cm^3^ for BNT-BT-Dy under *E* = 1750 kV/cm, noting that the breakdown field is not achieved because of the device limitation.

We have attempted to numerically adjust the curves obtained for the two films doped with the rare earth element using an exponential function, as expressed by Equation (6), which perfectly fits both curves:(6)$$y=Ae^{R_{0}x}+y_{0},$$

It is worth noting that in BNT-BT-Mn thin films, which are equivalent to the films investigated here, Yanjiang Xie et al. reported the possibility of applying a field of 2500 kV/cm reaching *W_rec_* = 54 J/cm^3^. We have taken this value as the maximum effective field that can be applied to our system. Replacing this value in Equation (6) allowed us to determine energy values of *W_rec_* = 12 and 16 J/cm^3^ for Nd and Dy films, respectively. The efficiency in this field has also been determined by linear fitting and was found equal to 58% and 64%, respectively.

### 3.4. Optical Investigations

Figure 9 shows the room-temperature emission spectra for the BNT-BT doped with a lanthanide element after excitation with a suitable wavelength. The emission spectrum of BNT-BT-Dy is shown in Figure 9a. The measurement was performed with a UV excitation wavelength of λ_ext_ = 360 nm. The obtained spectrum shows one intense emission at 573 nm, which corresponds to the ^4^F_9/2_ → ^6^H_13/2_ transition. Two other weak emissions were observed at 475 and 649 nm, which are attributed to the transitions ^4^F_9/2_ → ^6^H_15/2_ and ^4^F_9/2_ → ^6^H_11/2_ [33].

The BNT–BT–Nd sample was excited with 786 nm and emitted in the near-infrared region (Figure 9b). Three clear bands were observed at 875, 1057, and 1505 nm. The diode emission observed at around 900 nm corresponds to the transition of ^4^F_3/2_ → ^4^I_9/2_. It was attributed to the electronic transitions from the lowest ^4^F_3/2_ crystal field component to the ground manifold [34]. We noted the observation of only two ^4^I_9/2_ Stark levels instead of the expected five. The highest peak at 1057 nm was associated with the ^4^F_3/2_ → ^4^I_11/2_ electronic transition. A very small emission was detected at 1324 nm, which was attributed to the ^4^F_3/2_ → ^4^I_13/2_ transition. Finally, an intense band was observed at 1505 nm, which might be due to ^4^F_3/2_ → ^4^I_15/2_; such a transition is not usually observed and could be promising for Mid-IR lasing [35].

## 4. Conclusions

In summary, BNT-BT and BNT-BT-RE thin films were processed on a Pt/SiN substrate via chemical solution deposition. The incorporation of rare earth elements was found to improve the microstructure and decrease, by half, the average grain size, which considerably improved the electrical properties. X-ray diffractograms of the investigated films measured under grazing incidence mode confirmed the coexistence of both rhombohedral and tetragonal phases in all investigated films. The thermal variations in the dielectric constant revealed the presence of humps at around 120 °C for both BNT-BT and BNT-BT-Dy films, corresponding to the depolarization temperature. BNT-BT-Nd films exhibited one visible transition at 270 °C, which might be attributed to the complete *P4bm* transition. Such behavior is compatible with the highest MBP composition. All films showed an unsaturated hysteresis loop, characteristic of the complete ergodic relaxor phase. Moreover, films containing rare earth elements were found slimmer and endorse high energy storage properties comparable with values reported so far in other BNT-based thin films.

For BNT-BT, the recoverable energy was found to be around 4.5 J/cm^3^ with an efficiency of 45%. The inclusion of rare earth elements was found to be beneficial for increasing the breakdown field, on the occurrence of the storage properties. Under an effective field of 2500 kV/cm, the calculated recoverable energy and efficiency were found to be 12 J/cm^3^ (58%) and 16 J/cm^3^ (64%) for BNT-BT-Nd and BNT-BT-Dy, respectively. They also presented some attractive photoluminescence properties. The simultaneous appearance of ferroelectric and optical properties in the system under investigation is very promising for advanced optoelectronic devices.

## Figures and Tables

**Figure 1 materials-16-02197-f001:**
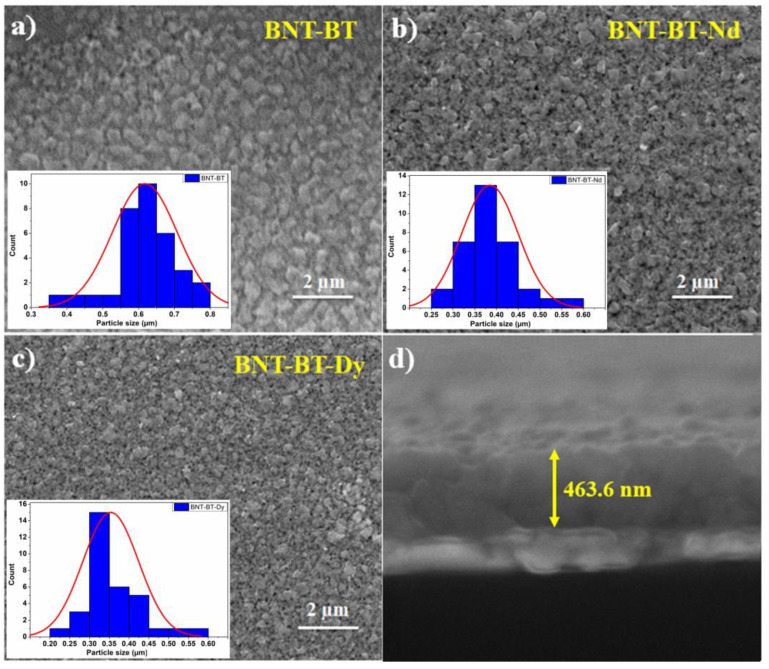
SEM images showing the microstructure of the investigated thin films: (**a**) BNT–BT, (**b**) BNT-BT-Nd, (**c**) BNT-BT-Dy, and (**d**) example of a cross-sectional image of the BNT-BT thin film.

**Figure 2 materials-16-02197-f002:**
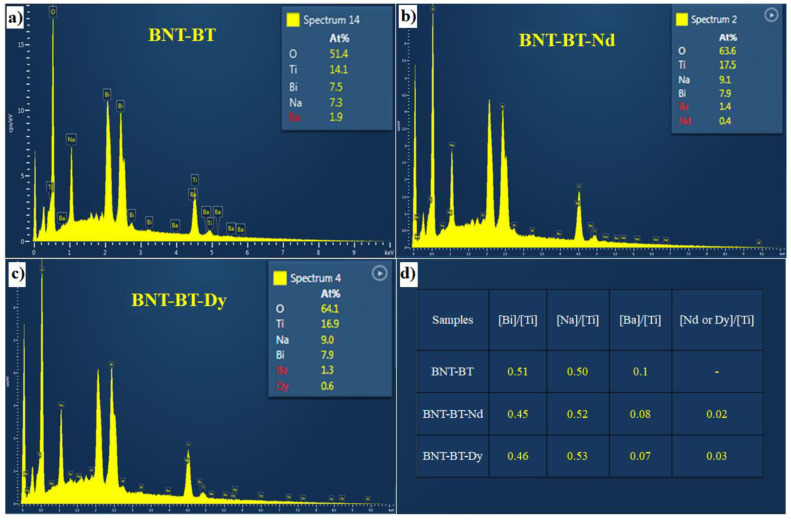
EDX spectra with the atomic percentage of (**a**) BNT-BT, (**b**) BNT-BT-Nd, (**c**) BNT-BT-Dy, and (**d**) table of atomic ratios.

**Figure 3 materials-16-02197-f003:**
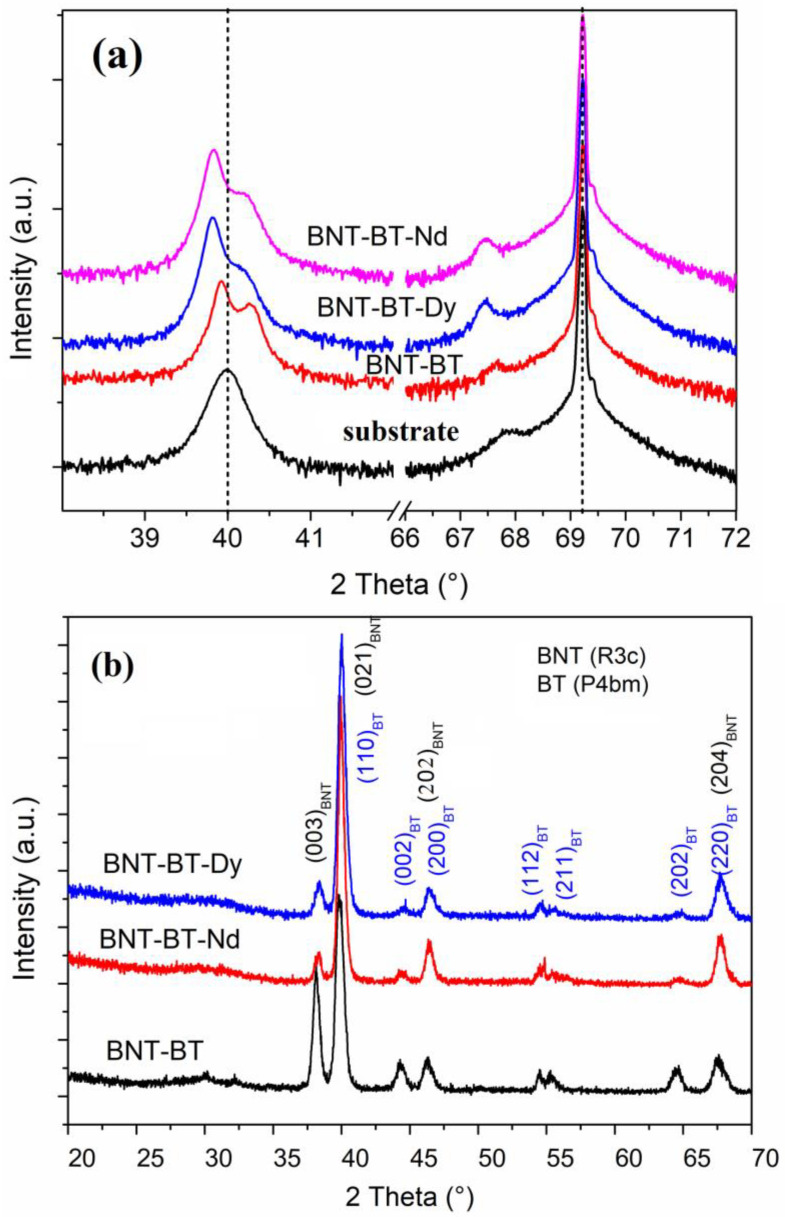
(**a**). Room-temperature X-ray diffractograms of BNT-BT, BNT-BT-RE (RE = Nd and Dy), and of the substrate, with a zoom of the X-ray diffraction patterns around 40° and 69°, and (**b**) X-ray diffractograms of the investigated films measured under grazing incidence mode.

**Figure 4 materials-16-02197-f004:**
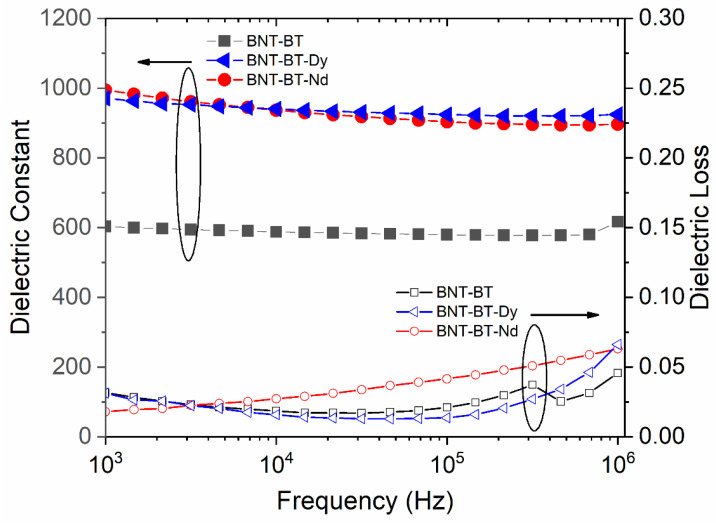
Room-temperature frequency dependence of the dielectric constants and the dielectric losses for the investigated samples.

**Figure 5 materials-16-02197-f005:**
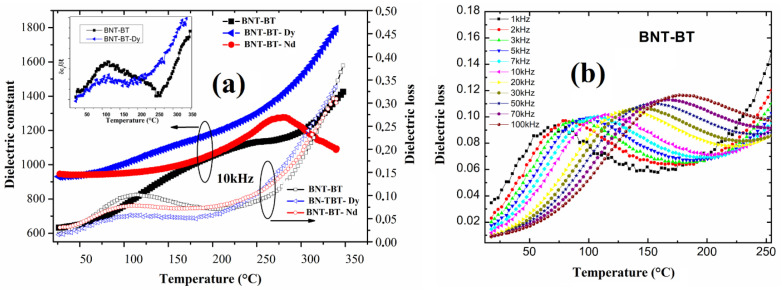
(**a**). Variation in the dielectric constant and the dielectric loss as a function of temperature for the investigated samples. The inset shows the derivative of the dielectric constant versus temperature for BNT-BT and BNT-BT-Dy compositions, and (**b**) an example of frequency dependence of the thermal variation in the dielectric loss for BNT-BT thin films.

**Figure 6 materials-16-02197-f006:**
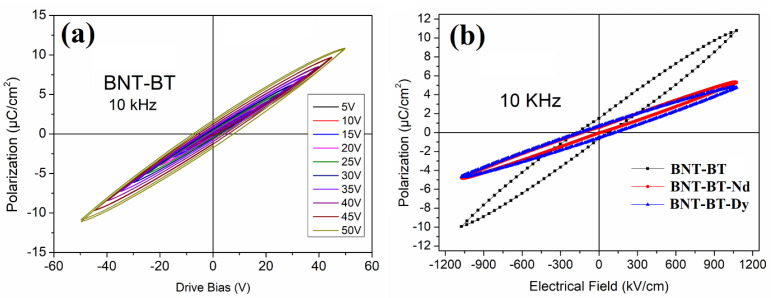
*P-E* hysteresis loops at room temperature for (**a**) BNT-BT thin films under different applied biases, and (**b**) the comparison of undoped and doped thin films at a 10 kHz test frequency.

**Figure 7 materials-16-02197-f007:**
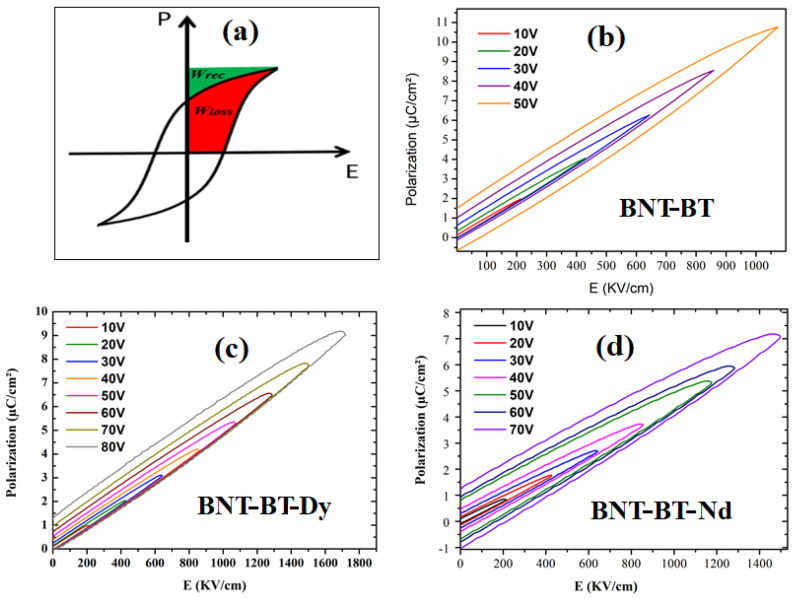
(**a**) Definition of the energy storage parameters; P-E hysteresis loops at room temperature in the positive part of the P-E loops: (**b**) BNT-BT, (**c**) BNT-BT-Dy, and (**d**) BNT-BT-Nd thin films.

**Figure 8 materials-16-02197-f008:**
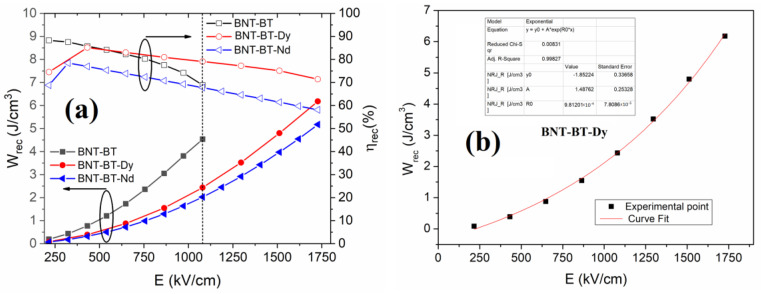
(**a**) Evolution of the energy storage parameters versus field density, and (**b**) Exponential fit of the experimental data for BNT-BT-Dy thin films.

**Figure 9 materials-16-02197-f009:**
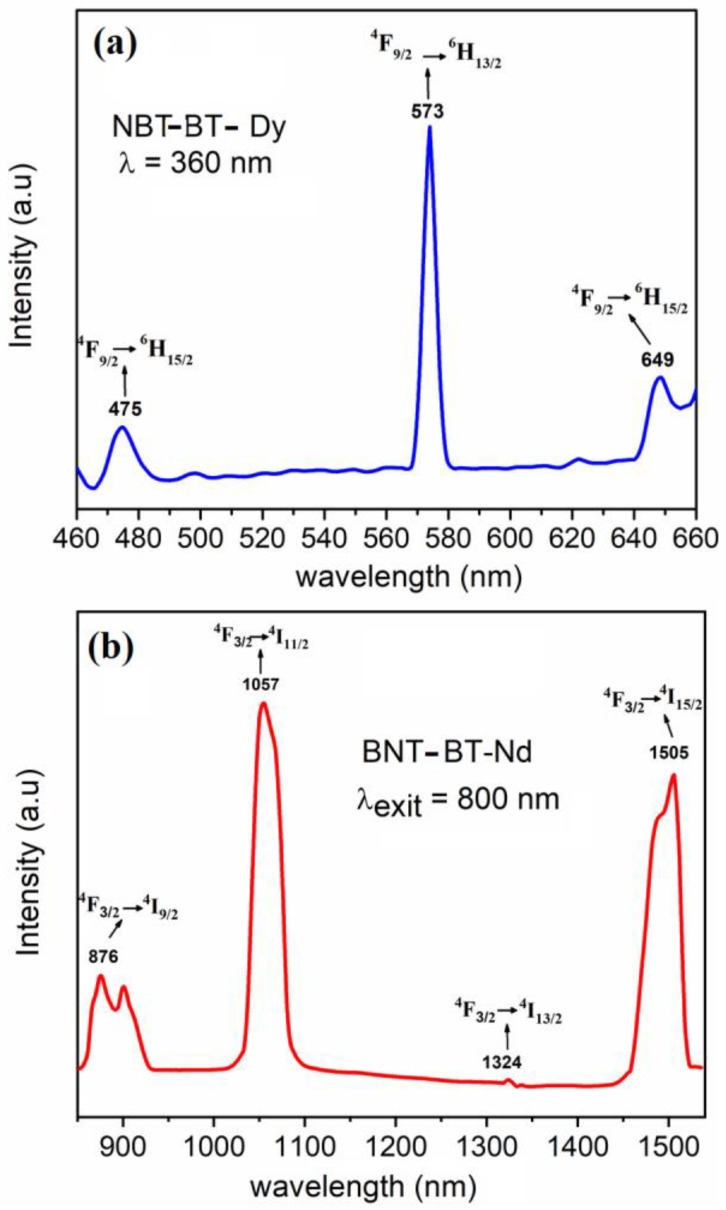
The emission spectrum of (**a**) BNT–BT–Dy after excitation at λ_ext_ = 360 nm, and (**b**) BNT–BT–Nd excited at λ_ext_ = 800 nm.

**Table 1 materials-16-02197-t001:** Comparison of the energy storage parameters of the investigated thin films with some reported bibliographic data on BNT-based thin films.

Samples	E_max_ (kV/cm)	W_rec_ (J/cm^3^)	η (%)	Ref.
BNT-BT	1080 (real)	4.5	45	This work
BNT-BT-xNd (x = 0.01)	1080 (real) 1780 (real)	2 5.3	68 59	This work
	2500 (effective)	12	58	
BNT-BT-xDy (x = 0.01)	1080 (real) 1780 (real)	2.5 6.6	79 72	This work
	2500 (effective)	16	64	
BNT-BT-xMn (x = 0.2)	2000	25	82	[7]
BNT-BT-xMn (x = 0.6)	2500	54	59	[7]
BNT-xHo (=0.02)	500	9.4	55	[2]
BNT-xEr (=0.02)	500	9.2	57	[2]
BNT-BT-NaNbO_3_ (=0.1)	3170	32	90	[9]
6BNT-4ST	10.4	64.5	64.5	[10]
0.94BNT-0.06BT with PLCT seed layers	17.2	74.3	74.3	[30]

## Data Availability

Not applicable.

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
