# Peer review of "Influence of the Addition of Rare Earth Elements on the Energy Storage and Optical Properties of Bi0.5Na0.5TiO3–0.06BaTiO3 Polycrystalline Thin Films"

_materials, 2023, doi:10.3390/ma16062197_

Round 1

Reviewer 1 Report

This is an interesting study related to improving the energy storage and optical properties of the lead-free perovskite materials (Bi0.5Na0.5TiO3 -0.06BaTiO3) by dopping rare earth elements, Neodymium (Nd) and Dysprosium (Dy). However, this study should be significantly revised before publication according to following major problems.

1.In Figure 1, SEM element mapping must be inserted to confirm the sucessful dopping of Nd and Dy. Moreover, Figure 1d is poor, please provide a image with higher quality 

2. In the introduction section, the authors should emphasize the current problems or disadvanatges of pure BNT perovskites. This demonstrates necessary of this study as well as the dopping process.

3. XRD parterns of substrate, BNT, and dopped BNT were not different. I suggest that the authors should re-characterize them.

4. All abriviations should be presented along with a full description in the first appearence

Author Response

Reply to the reviewer's comments:

Reviewer 2 Report

In the present manuscript, the authors present a detailed study of rare earth element doped  BiNaTiO-BaTiO polycrystalline thin film.

In general, I think indeed the author conduct various of experiments to characterize film properties. However, I still have some concerns about the novelty/motivation of the current manuscript and some technical details mostly about the figures.

I therefore cannot recommend its publication in the current form since a major revision is required to meet the standard of materials.

Below, I describe all this in detail as well as specify the points to address to improve the consistency/clarity of the manuscript.

Comment 6: In resistance measurement, does the thickness of the nanoparticle films taken into account?

Major:

comment 1: My most concern would be what is the motivation for all the experiments that were conducted. Showing figures one by one in a roll looks more like an experimental technique report other than a scientific manuscript. Lots of information/conclusions the author extracted are fairly vague or uncertain.

comment 2: Word usage could be improved since it's for scientific reports other than general intro-report. The author should address this issue accordingly. (line 107...etc.)

comment 3: In Figure 2, at 68 degrees, there is no peak splitting/shifting observable. The author should elaborate on the reason.

comment 4: In Figure 2, this is a series of peaks at around 64 degrees. Has the author looked into that? Where is that signal come from? Same issue for those of around 78 degrees.

comment 5: Line 112-115, what is the evidence or signature that can possibly prove such assumption?

comment 6: Line 125, does this mean that the reason for the dielectric constant increasing is unclear or lack of understanding?

comment 7: Figure 4a, does -Dy sample also have a similar transition?

Minor:

comment 1: Page 1, line 31, citation marks should be combined.

comment 2: check the usage of "may" and "can", for instance, in line 137, "as can be shown..." is a bit strange.

Author Response

Reply to the reviewer's comments:

Reviewer 3 Report

1. The author can add to the introduction section other lead-free perovskite-structured compounds that exhibit potential dielectric, ferroelectric, and magnetic properties. Some related works may be cited in the revision to extend the readership, DOI: https://pubs.acs.org/doi/10.1021/acs.inorgchem.1c03624 https://doi.org/10.1021/acs.chemmater.5b03564 https://doi.org/10.1016/j.nanoen.2016.02.036 ; etc. 

2. The XRD analysis results in Figure 2a show a very weak diffraction pattern of the BNT sample and do not even appear as the main peak 117 at ~35°, although Figure 2b shows a splitting of the BNT peak and the substrate. The author must explain this phenomenon in the discussion. The XRD pattern in Figure 2 must be added a symbol on the peak of the BNT compound.

3. It is preferable to show the particle size range in the SEM section for all samples, especially since the author explained this size decreases in the presence of rare-earth doping. The author also shows the thickness of the layer for the BNT sample, please enter the thickness value for the doped sample.

4. The dielectric analysis shows an increase in the transition temperature and a constant dielectric value (Figure 3 and 4), but the ferroelectric analysis shows a decreasing polarization value with RE-doped, why is this trend the opposite? The dielectric value should also show that the electric current is restrained so that the polarization value also increases, please discuss this in the manuscript.

5.  Table 1 shows the storage efficiency values of all samples. However, as seen in Figure 7a, the BNT-BT efficiency value in the table does not match the value in the figure drawn at 1080 kV/cm. Please check again and revise the values in table 1 according to figure 7. then I suggest in Table 1 to compare the wrec and efficiency values at the same electric field value at 1080 kv/cm for the Nd and Dy samples other than at a maximum value of 1780.

6. The conclusion needs to be improved by including the optimum results of energy storage parameters from the best ceramic sample

 Some minor corrections in the manuscript:

·      Typo error in the title.

·      The sample code in Figure 1b and 1c, sem image should be changed using strips (-)

·      the layer thickness value in image 1d is not clear, please provide a higher resolution

·      Writing the Wrec symbol needs to use subscript, please check all the writing with superscript or subscript.

Author Response

Reply to the reviewer's comments:

Round 2

Reviewer 1 Report

It can be accepted for publication

Author Response

Thank you very much for your positive consideration of the revised version. 

Sincerely yours,

The corresponding Author

Reviewer 2 Report

The author has made significant effect in the revised version. I recommend its publications in current form.

Author Response

(The authors gave the same response as above.)

Reviewer 3 Report

I am satisfied with the fundamental explanation and correction given by the author regarding the comments, I think the present version is acceptable in the material journal.

However, before being accepted, the author must pay close attention and revise again Table 1.

* The author has added the Wrec and efficiency values at 1080 kV/cm for all samples based on Figure 8. However, when compared again, the efficiency values of the three samples are incorrect, the values refer to the right side of the Y axis, and the three graphs are at the top. Check this value again, it should be a efficiency value for BNT-BT around 69%, BNT-BT-Dy around 79% and BNT-BT-Nd around 68% (Check the actual value). The values listed in Table 1 in my opinion refer to the bottom of the graph which is the Wrec graph. Recheck the values of Table 1.

*The sample name in graph 8a also needs to be revised, it is written BBNT-BT-Dy on the graph which should be BNT-BT-Dy

Author Response

We would like to thank reviewer for the careful reading of the revised manuscript.

Sincerely yours,

The corresponding Author
